# Grass Pea (*Lathyrus sativus* L.)—A Sustainable and Resilient Answer to Climate Challenges

Letice Gonçalves [1,*], Diego Rubiales [2], Maria R. Bronze [1,3,4] and Maria C. Vaz Patto [1]

1  ITQB NOVA—Instituto de Tecnologia Química e Biológica António Xavier, Universidade Nova de Lisboa, Av. da República, 2780-157 Oeiras, Portugal; mbronze@ibet.pt (M.R.B.); cpatto@itqb.unl.pt (M.C.V.P.)
2  IAS—Institute for Sustainable Agriculture, CSIC, Av. da Menéndez Pidal s/n, 14004 Cordoba, Spain; diego.rubiales@ias.csic.es
3  FFULisboa—Faculdade de Farmácia da Universidade de Lisboa, Av. das Forças Armadas, 1649-019 Lisboa, Portugal
4  IBET—Instituto de Biologia Experimental e Tecnológica, Av. da República, Estação Agronómica Nacional, 2780-157 Oeiras, Portugal
*  Correspondence: leticeg@itqb.unl.pt

**Abstract:** Grass pea (*Lathyrus sativus* L.) is an annual cool-season grain legume widely cultivated in South Asia, Sub-Saharan Africa, and in the Mediterranean region. It is a stress-resilient crop with high nutritional value, considered a promising source of traits to breed for adaptation/mitigation of climate change effects. It is also reported as a suitable crop for more sustainable production systems such as intercropping. In this review, we elaborate an integrative perspective including not only an agronomic-based but also a variety-breeding-based strategy in grass pea to deal with climate change impacts, summarizing the current knowledge on grass pea biotic/abiotic stress resistance. Additionally, we highlight the importance of implementing fundamental techniques to create diversity (as interspecific hybridization or gene editing) and increase genetic gains (as speed breeding or the efficient identification of breeding targets via genomics) in the development of multiple stress-resistant varieties that simultaneously provide yield and quality stability under climate vulnerable environments.

**Keywords:** grass pea; *Lathyrus sativus*; climate change; genotype x environmental interaction; genome-wide association studies (GWAS); healthy food; speed-breeding; hybridization; intercropping

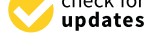



## 1. Introduction

In a world facing climate change and associated environmental stresses that hamper agricultural productivity and food security, the requirement for more sustainable agriculture is on the rise. Climate change, defined as "any change in climate over time whether due to natural variability or as a result of human activity" [1], has limited agricultural productive growth by 21% over the past 60 years [2]. This growth limitation coupled with both the stagnation of yield breeding gains and increasing ecological pressure for production systems inputs reduction, constitutes a huge challenge for both farmer and the research communities [3]. These communities are well aware of the urgency to obtain more environmentally sensitive agricultural practices to enable the mitigation of climate change impacts, and more resilient crops and varieties to adapt to and tackle climate change [4].

The increasing demand for environmental-friendly agricultural practices and food security establishes a favourable context for new cropping systems that include grain and forage legumes [5,6]. Grain legumes, also known as pulses, are major foodstuffs and important sources of protein in most countries [7]. They are also environmentally friendly sustainable sources of many other nutritional and health-beneficial components [8,9].

Grass pea (*Lathyrus sativus* L.) is an annual cool-season grain legume crop, that due to its relatively low input requirements compared to major crops, is considered a model

crop for sustainable agriculture and an interesting alternative for cropping systems diversification in marginal lands [10,11]. It is characterized by a wide adaptation to different soils and climates, to low temperatures, showing flood and drought tolerance, insect and disease resistance, and high protein content for human and animal feed [12,13]. Moreover, it is superior in yield, nitrogen fixation, and salinity tolerance, when compared to other legume crops [11]. These traits make it an outstanding crop for ensuring nutritional security, especially in the face of impending climate challenges [14]. As an example, the importance of grass pea was recognized by Kew's Millennium Seed Bank which considered it among the priority crops to be used for the adaptation of the world's most important food crops to new climatic conditions production [15,16].

Recent reviews [14,16–18] highlighted grass pea as a stress-resilient crop with high nutritional value, contributing to a better health state and capable to withstand climate change impacts. Additionally, the progress of its improvement by conventional breeding until the more recent genomics techniques was also highlighted in those works.

In the present review, under the scope of mitigation and/or adaptation to climate change impacts, we elaborated a more integrative perspective that includes both agronomic-quality-based and variety-improvement-based strategies (Figure 1). We addressed climate change as an opportunity for grass pea expansion into new regions and grass pea as a source of important resilient traits capable to withstand environmental stresses, detailing the state-of-the-arts on the genotype by environment interaction on agronomic and quality aspects. As a model crop for sustainable agriculture, we discussed the potential of agroecological transition practices such as intercropping. Finally, we focused on breeding highlighting techniques that allow shortening the time of selection cycles, and genetic studies, such as genome-wide association, to clarify the putative candidate genes and mechanisms underlying interesting traits, contributing at the end also to increase the efficiency of breeding against climate change.

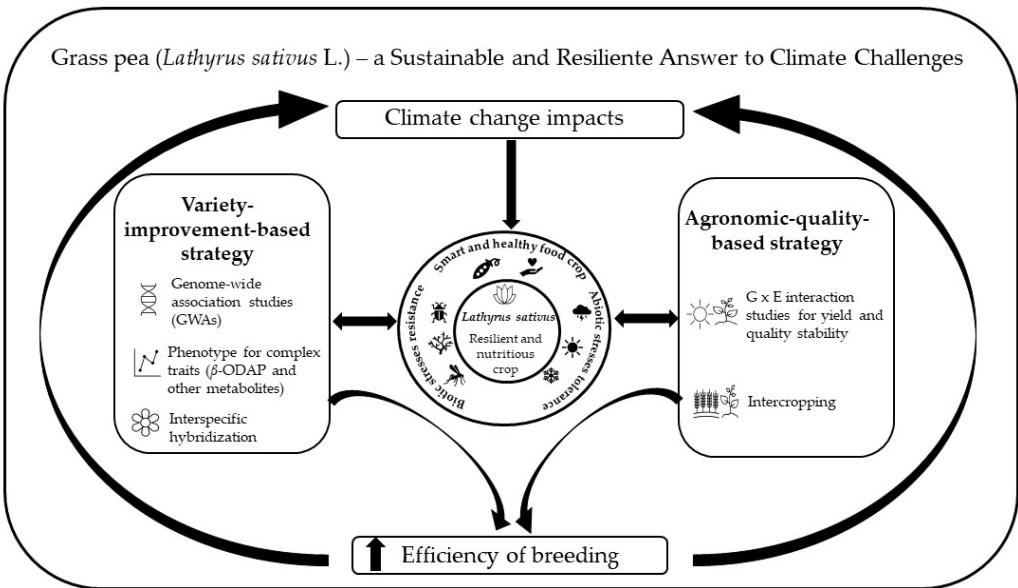

**Figure 1.** Integrative perspective that includes both agronomic-quality-based and variety-improvement-based strategies regarding grass pea improvement toward an efficient answer to climate change.

## 2. New Challenges and Opportunities Due to the Impacts of Climate Change in Grass Pea Production

Although climate change has been a constant process on earth, for the last century the pace of the variations has become more frequent [19] posing increasing constraints to crop production and agricultural systems. The most significant stresses are and will be due to variable rainfall, reduced water availability, temperature raising, and more frequent periods

of extreme temperatures that will have major implications for the geographic distribution of crops [3,20]. Moreover, also the spread and intensity of pest and disease outbreaks and weed expansion into higher latitudes or altitudes are and will be influenced by this temperature and rainfall variability, with a strong impact on agricultural yield and crop management [3,21,22]. The impact of these stresses in agriculture will result in a decline in crop yields, and thus alternative crops or new varieties are required to ensure a stable food supply [21].

Many of the regions cultivating legumes as staple sources of plant proteins rely on rain fed systems for crop growth with limited access to resources such as irrigation or fertilizers [14]. Within these regions, optimal temperatures must range between 15–25 °C, with a base temperature of 0 °C, for cool-season legumes, and between 25–35 °C, with a base temperature of 10 °C, for warm-season tropical legumes [23]. As stated previously, grass pea is a cool-season legume crop, and due to climate change, the current grass pea production regions are facing increases in frequency and severity of extreme weather events [14], which could lead to a shift in cropping seasons [21,24]. All these predicted scenarios pose both new challenges for grass pea in traditional production regions, but also opportunities for expanding into new areas.

Grass pea is originated from southwest and central Asia, subsequently spreading into the eastern Mediterranean [25]. Worldwide, grass pea is regarded as a major crop in Bangladesh, India, Nepal, Pakistan, and Ethiopia, and cultivated to a lesser extent in many European countries (from south Germany to Portugal and Spain and east to the Balkans and Russia), the Middle East (Syria, Lebanon, Palestine, Iraq, Afghanistan), Northern Africa (Egypt, Morocco, Algeria), China, Chile and Brazil [26–28]. Grass pea production has decreased in the Mediterranean [29,30] but increased in Bangladesh and Ethiopia. Indeed, in Ethiopia, grass pea has recently ranked 19th on the 21 highest priority crop species, and in South Asia ranked 22nd of the top 24 [31]. This increase in production could be the result of the recent attention that grass pea has received for cultivation in problematic soils and new niches like rice-fallow [14,32].

Three important Asian grass pea-producing countries, namely India, Bangladesh, and Nepal, will have to deal with different challenges of climate change, but in particular with floods [31,33]. Due to the expected increase of the summer Asian monsoon rainfall, rising sea levels will contribute to increasing coastal flooding in low-lying areas [34,35]. Grass pea flood-tolerant varieties will be extremely important for future production under these conditions. In addition, water scarcity, salinity, and rising temperatures are predicted to be a concern, mainly in the west coast and southern India, and northern Bangladesh, with the particularity of high temperatures heavily influencing the changing scenario of pests and diseases [24,36]. Grass pea varieties with multiple biotic and abiotic stress resistance will be needed to adapt to these developing production constraints.

In Africa, mainly in the Mediterranean regions, a similar pattern of increased temperature is expected, and extreme heat events will occur with more frequency and intensity. Considering the projected mean precipitation decreases [34], it is expected that parts of those regions could become drier, leading to increasing desertification [37]. In Europe and in particular, in the Southern regions, where a "Mediterranean" climate prevails, grass pea cultivation has suffered already a severe reduction during the last century [29,38]. However, there is a renewed interest to re-introduce this hardish crop into Mediterranean rainfed cropping systems where it can be an alternative to overcome the expected climate change impacts [11,39]. The "Mediterranean" climate is characterized by mild wet winters and warm to hot dry summers, with the annual rainfall occurring during the winter half-year [40]. With this peculiar rainfall distribution, severe water deficit can commonly occur during the cool-season legumes growing season, even when there is sufficient annual precipitation since crop yield is mostly dependent on changes in seasonal cycles of precipitation rather than on variation in the annual average value [41]. Due to climate change, annual precipitation is expected to decline over much of the Mediterranean region south of 40–45° N [42], where cool-season legumes are traditionally grown. Consequently, drought

is predicted to occur 10 times more frequently in the future over a large part of those regions [43], hampering grain legumes local cultivation. In addition, an increase in average temperatures is predicted in the whole Europe, with Southern regions suffering from the increase in the frequency of extreme heat, while Northern Europe becoming warmer.

Although this might hamper the more traditional southern grass pea production, a northern expansion of its actual under cultivated area might become a possibility [11,28,44,45].

## 3. Grass Pea as a Source of Important Traits to Tackle Climate Change

Grass pea is a hardish crop with reported tolerance to extreme temperatures, drought, flooding, and salinity being able to grow successfully in warm climates, and marginal and nutrient-deficient soils, delivering reasonably good yield despite unfavourable growing conditions [12,32,33,46]. Moreover, grass pea is resistant to many diseases and pests, compared to other legume crops [11,27]. Therefore, it is a promising source of traits to breed for adaptation to climate change, not only for its own varietal breeding but also for the development of more adapted varieties of related major legume crops, such as peas [47].

### 3.1. Grass Pea Abiotic Stresses Tolerance

One of the major factors impairing crop growth and yield is water deficit [48]. Since climate change might lead to variable rainfall and overall reduced water availability, imposing drought stress, the main focus of improving plants' resilience to climate change should rely on strategies that promote both, saving water and improving water capture efficiency.

Saving water can be achieved by improving plants' water use efficiency (WUE) and/or developing morphological drought tolerance traits as adaptive mechanisms [49,50]. These mechanisms represent three drought-adaptation strategies: escape, where the crop completes its life cycle before the onset of terminal drought; avoidance, where the crop maximizes its water uptake and minimizes its water loss; and tolerance, where the crop continues to grow and function at reduced water content [51,52]. Grass pea seems to be, mainly, a drought-tolerant and/or avoidance crop [53]. This feature was observed in grass pea with delayed maturity and senescence in Mediterranean-type environments with a short growing season and terminal drought [51,53]. Although water deficit can decrease yield due to flower and pod abortion, seed size seems not to be affected in grass pea [29,54]. Seed size consistency in response to water deficit can be a useful tolerance adaptation of grass pea to drought stress. Likewise, grass pea winged and narrow leaves, able to roll inward of leaf margins to diminish water loss, constitute a drought avoidance strategy [16,55]. Additionally, escape mechanisms such as early maturity, early vigour, and early flowering can also be found in grass pea [56]. This would be of major importance to yield in short-season environments, such as the Mediterranean ones [53].

Another way to improve the resilience of plants to climate change impacts is by improving water capture efficiency through for instance a deep rooting system, as it allows access to unexploited water resources when the soil surface desiccates [57]. Indeed, grass pea has a hardy and penetrating root system [58], whose exoproteome revealed abundant number of proteins responsive to abiotic and biotic stresses [59], suited to a wide range of soil types [46]. It has been hypothesized that this could be the basis for its considerable drought but also flood tolerance [27]. Indeed, during flooding periods, the lower soil layers remain aerated, and thereby a penetrating root system will allow grass pea to escape from flood constraints [14].

Grass pea flooding tolerance is highlighted by Dixit et al. [26] and by Girma and Korbu [46], especially during the germination phase, which in Asia, is exploited by broadcast grass pea in the rice crop in the wet soil, 4–5 weeks before rice harvest [26,60]. These features could be quite interesting to tackle flooding events that are predicted to occur more frequently also in other regions, such as the Mediterranean.

One of the main causes of soil degradation in the world is salinity [61]. Salinity reduces osmotic potential making it more difficult for the plant to extract water, increasing surface crusting, impairing water infiltration, and reducing root zone aeration thereby affecting

plant growth and reducing crop yield [61,62]. Arid and semi-arid regions are the most prone to desertification and salinization [62], but also in coastal regions, such as Bangladesh, an important region of grass pea production, a rise in sea level is causing seawater intrusion and consequently soil salinization. Soil salinization has a significant cost for plants; hence, to minimize the impact of salinity the way forward is to breed greater salt tolerance into present crops and to introduce new species for cultivation [61]. Grass pea capacity to withstand moderate salinity has been recognized [11]. This grass pea capacity may be due to a salinity tolerance mechanism resulting from increased activity of the antioxidant system and efficient compartmentation of harmful ions in the roots and shoots [63–65]. These works bring new insights into the grass pea tolerance to salinity and position it in the front line as a priority crop to face salinity stress.

High temperatures affect legume crops in several states of development. A daily maximum temperature above 25 °C is considered the upper threshold for heat stress in cool-season crops. The impact of heat stress depends on the intensity, duration of exposure, and the degree of the elevated temperature [66]. High temperature (>30 °C) during flowering reduces pollen viability, increases flower drop, and reduces seed set/pod filling, thereby limiting grain yield [14]. The phenology of crops in earlier reproductive phases is critical for escaping environmental constraints such as heat stress [67]. Grass pea is a cool-season legume and therefore could be affected by high temperature although it shows tolerance to heat [14]. Kumar and Tripathi [68] who conducted a study to analyse the effects of temperature factors on *L. sativus* highlighted this heat tolerance. In this study, grass pea seeds of an $F_1$ generation were exposed to 55 °C for 48 h and after morphological and cytological analysis of $F_2$ generation plants, the seed production was not affected by the heat stress and the sterility was not too high to affect the fertility of the grass pea plants.

Another grass pea mechanism to escape anthers and stigmas desiccation by heat and/or wind during the flowering period are cleistogamous flowers, promoting good seed-set during variable weather conditions [56,69]. The flowering period can also be anticipated, as stated previously for drought response, then precocity is considered also an important trait to avoid terminal heat stresses, especially during pod-filling [56].

### 3.2. Grass Pea Biotic Stresses Resistance

Due to climate change, some diseases tend to move their area of action from one region to another. An important strategy to deal with pests and diseases outbreaks could be the use of resistant cultivars that are considered the safest, most economical, and most effective crop protection method in disease prevention [70]. Compared to other legumes, grass pea is resistant to many diseases and pests [71].

Powdery mildew is among the major diseases that affect *L. sativus* [12]. However, resistance to powdery mildew (*Erysiphe pisi* and *E. trifolii*) has been reported in *L. sativus* [12,72–74]. Quantitative resistance to *E. pisi*, due to resistance to epidermal host cell penetration and not associated with host cell necrosis, was described in *L. sativus* by Vaz Patto et al. [73]. In that study, diverse levels of resistance were detected both in growth chambers at seedling stage and especially under field conditions at adult plant stage [73]. Recently, Martins et al. [74] observed in a worldwide germplasm collection of 189 *L. sativus* accessions, a wide range of responses, with partial resistance to *E. trifolii*, previously uncharacterized, being less frequent, compared to *E. pisi*. Furthermore, these authors performed a genome-wide association study (GWAS) on the grass pea interaction with *E. pisi* and *E. trifolii* and identified 7 and 12 different single nucleotide polymorphic molecular markers (SNPs) associated with *E. trifolii* and *E. pisi* responses respectively, anticipating that the oligogenic resistance to both pathogens has a different genetic basis. The SNP-trait association common to both pathogens was located in a gene encoding for Ogre retrotransposons. Other candidate genes proposed were putatively involved in gene expression regulation or coded for an NB-ARC domain.

Rust resistance has also been identified in *L. sativus* germplasm [75]. A transcriptome analysis conducted by Almeida et al. [76] in two *L. sativus* contrasting genotypes, inoculated

with rust, provided a comprehensive insight into the molecular mechanisms underlying pre-haustorial rust resistance in grass pea. Fifty-one genes were identified as potential resistance genes, prioritizing them as specific targets for future functional studies on grass pea/rust interactions. More recently, Martins et al. [77] identified new promising sources of partial resistance to rust in the previously mentioned worldwide *L. sativus* collection of accessions, under controlled conditions, and its genetic architecture and mechanisms have been clarified through GWAS. Seven different grass pea genomic regions were detected significantly associated with *U. pisi* disease severity, suggesting that the observed partial resistance is oligogenic. Candidate genes proposed encoded for leucine-rich repeat and NB-ARC domain, and TGA transcription factor family.

Ascochyta blight infection on grass pea has commonly been attributed to *Ascochyta pinodes* (telomorph *Dydimella pinodes*) [78], and only recently also to *Ascochyta lentis* var. *lathyri* [79]. Unfortunately, no resistance screenings have been reported so far using these new *A. lentis* var. *lathyri* isolates. *A. lentis* var. *lathyri* is very specific infecting grass pea only, whereas *A. lentis* isolate from lentils could infect grass pea but at low levels. Similarly, cross inoculations studies showed that grass pea accessions can be very susceptible to *A. pinodes*, but are immune or highly resistant *A. rabiei*, *A. lentil*, and *A. fabae* isolates [78]. *Ascochyta lathyri* has been reported in other *Lathyrus* species, but not on grass pea [80]. Nevertheless, resistance to *A. pinodes* has been recorded on accessions of *L. sativus* [80–82]. This resistance is of major interest considering the possibility to be transferred to the phylogenetically related field pea (*Pisum sativum*) crop. Ascochyta blight is a major constraint to the production of field pea and complete resistance to the infection has not been observed on this species [81].

Soil-borne diseases, such as fusarium wilt have a tremendous impact on a wide range of plant species, including grass pea. Nevertheless, resistance ranging from high to partial was described in grass pea germplasm against *Fusarium oxysporum* f. sp. *pisi*, and its genetic architecture and mechanisms have been clarified [70,83]. In total, 17 genomic regions were associated through GWAS with three fusarium wilt response related traits in grass pea, anticipating an oligogenic control, and candidate genes proposed were involved in secondary and amino acid metabolism, RNA (regulation of transcription), transport, and development [83]. Another soil-born constrain for grass pea cultivation is the root parasitic weed crenate broomrape (*Orobanche crenata*) [30,84]. The *L. sativus* species is susceptible to this parasite and, until the moment, no real resistance was identified. However, precocity could be considered as avoidance to crenate broomrape. Rubiales et al. [39] recommended early grass pea cultivars for areas prone to high broomrape infection, whereas cultivars with a longer growth cycle are more suitable to environments with low or moderate broomrape incidence.

Beyond diseases, insect pest outbreaks and changes in their distribution due to climate change are major concerns due to their negative impact on crops yield. Compared to other legume crops, grass pea shows resistance to many pests including storage insects [27]. The most serious grass pea pests in India, Bangladesh, Ethiopia, and Nepal are thrips (*Caliothrips indicus*), aphids (e.g., *Aphis craccivora*), and pod borers (*Etiala jhinkinella*) [85]. Fortunately, several authors, as reviewed by Vaz Patto et al. [11] described some resistance to pests in *L. sativus*. Infestation with *Bruchus pisorum* represents an added challenge since it can destroy large portions of the stored grass pea harvest before the next crop [86], and until the moment no significant resistance was found.

Under the pressure imposed by climate change, the development of varieties displaying resilience to prevalent biotic and abiotic stresses has gained new strength. Indeed, grass pea presents a set of important resilient traits. Thus, may become a donor for genes expressing adaptive traits such as disease resistance, drought, flooding, salinity, and heat [47]. The identification of sources of these adaptive traits is essential, with pre-breeding efforts being paramount for this task, as well as the understanding of their genetic control that will allow pinpointing promising genomic targets for the development of molecular tools to assist stress resistance precision breeding in this species.

Nevertheless, climate change is not only highlighting the importance of the development of multi-stress resistant varieties but is also calling our attention to the potential negative impacts that might directly or indirectly impose on the quality of crops.

## 4. Yield vs. Quality Stability in Grass Pea

Besides being a model crop for sustainable agriculture, grass pea provides food and nutrition security to many low-income communities, being a highly nutritive food crop [30]. Despite those advantages, grass pea is still an underused crop due to its low yields but also its content on the neuroexcitatory $\beta$-N-oxalyl-l-$\alpha$,$\beta$-diaminopropionic acid ($\beta$-ODAP) considered the cause of the neurodegenerative disease—lathyrism, if consumed as a staple food for extended periods of time [33,47,87]. Since the identification of $\beta$-ODAP in grass pea in 1964 [88], this harsh and resilient crop suffered from a reputation of being toxic. However, under an equilibrate diet, including cereals and fruits, lathyrism can be prevented, and grass pea can be safely consumed [16,30,89].

Taking the above in consideration, grass pea breeding has focused mainly on enhancing yield and yield stability as well as on producing seeds with high nutritional value, meaning high protein and reduced $\beta$-ODAP content [29,90,91]. Both yield and quality are complex traits [30,92]. Indeed, it seems that also the $\beta$-ODAP content as well as yield are highly influenced by climatic and edaphic conditions and display a high genotype-by-environment interaction (G × E) [93,94]. Clear G × E interactions are frequently identified on the metabolomics profiles of grain legumes, as can be the case of secondary metabolites exerting functions related to environmental conditions' adaptability, such as defence against abiotic stresses like heat stress [95]. Understanding G × E interaction is one of the most important steps in a breeding program to match genotypes and environments in such a way that optimal genotypes are selected [96,97]. Moreover, a breeding program aims to provide farmers with genotypes with guaranteed superior performance, and this can be achieved also by exploiting its local or broad adaptation [29]. Although some grass pea G × E interaction studies have been conducted [29,30,91,98], the available data is still scarce, and more research must be promoted to a better understanding of the genetic and environmental factors leading to an optimal phenotype [97].

### 4.1. Grass Pea, a Smart and Healthy Food Crop

Grass pea is considered a smart and healthy food crop, being valued and cultivated for its high protein content in seeds [16,99]. The seed of *L. sativus* has high amounts of protein, low fat, and high starch content. Grass pea protein content (18–34% in seeds and in mature leaves (17%), is higher than field pea (*P. sativum*) or faba bean (*Vicia faba*), but lower than soybean (*Glyxine max*) [13]. Grass pea proteins, mainly composed of globulins, albumins, and glutelins, are rich in amino acids such as lysine but usually poor in sulphur-rich methionine and cysteine amino acids [16,100]. Besides that, grass pea is rich in *L*-homoarginine, a nonprotein amino acid present in concentrations up to 1% of the dry weight [86]. Indeed, it is the only known dietary source of *L*-homoarginine, an alternative substrate for nitric oxide biosynthesis, with advantages in cardiovascular physiology and general wellbeing. A daily intake of *L. sativus* as part of a normal diet could provide enough of this healthy compound [16,101]. Moreover, *L*-homoarginine is also associated with benefits in overcoming the consequences of hypoxia associated with cancer tumour development [102].

Fikre et al. [100] found that, in grass pea, glutamic acid is usually present at high concentrations (0.03–0.08%), followed by aspartic acid (0.01–0.04%), arginine (0.01–0.05%), and asparagine (0.03–0.15%) in a similar pattern as for soybeans and lentils. Additionally, Grela et al. [99] found that grass pea seeds are rich in potassium (9.8 g kg$^{-1}$ DM) and several minerals such as copper, zinc, iron, and manganese for which average levels were 5.1, 44.1, 62.1, and 23.7 mg kg$^{-1}$ DM, respectively. Furthermore, grass pea is an interesting source of health-beneficial dietary lipids, with a high polyunsaturated fatty acid proportion

(58%) and phenolic compounds with high antioxidant activity, such as an average value of 68 mg/100 g of Gallic acid [8,90,99,103].

*4.2. The Influence of the Environment on Quality-Related Traits with a Putative Rule in Grass Pea Resilience*

Trait phenotypic expression depends on factors such as the genotype (G), the environment (E), and the genotype by environment (G × E) interaction. G × E interaction results from the differential expression of genotype (G) over the environment (E), which hampers the genotype selection for a target trait when the selection is meant for across a range of environments [104]. Nevertheless, due to climate change, directional selection for adaptation to changes in the environment will be required [105]. Therefore, the understanding of the G × E interaction on the traits under selection will be crucial for its better exploitation in breeding.

Sellami et al. [90] described a significant G × E interaction for some quality-related traits such as phenolic compounds and antioxidant activity. A significant G × E interaction for *β*-ODAP content was reported in some studies [46,93,94,106]. On the contrary, authors such as Hanbury et al. [98] and Chatterjee et al. [91] found no significant G × E interaction effect for *β*-ODAP. A careful analysis of the G × E interaction effect for *β*-ODAP is of extreme importance since grass pea varieties with low *β*-ODAP content are particularly important for arid and semi-arid developing countries, where grass pea is still a staple food, and its toxic risk must be reduced as much as possible. However, these may also be the regions that due to particular stressful climatic conditions (drought), *β*-ODAP content may increase. It has been established that some proteins and metabolites are generated in different tissues of crop plants in response to environmental biotic and abiotic stresses [67]. Under stress conditions, namely drought and salinity, grass pea tend to synthesize a set of metabolites, such as phenolic compounds, soluble sugars, proline and peroxidases, for instance [65,107,108]. Particular metabolites, such as soluble sugars, proline, abscisic acid (ABA) and *β*-ODAP appear to be correlated with grass pea drought and salinity stresses resilience [109,110]. Tokarz et al. [65] highlighted the role of these metabolites as osmoprotectants. Osmoprotectants have an important role in the osmotic potential adjustment [55,109–111], a plant's mechanism for drought tolerance [55,112]. Xing et al. [112] had already described the role of *β*-ODAP as osmoprotectant when analysing the relationship between the accumulation of *β*-ODAP and water stress, suggesting that the content of this metabolite increased as the drought tolerance of the grass pea variety increased. Girma and Korbu [46] and Jiao et al. [94], described the increased production of *β*-ODAP because of drought, zinc depletion, and excess of iron or cadmium in the soil. A recent transcriptomic study conducted by Verma et al. [113] suggested a differential expression of several stress-related and hormone-related genes, upon PEG stress, in two contrasting grass pea cultivars, Pusa-24 which has five to six times higher *β*-ODAP content than Ratan. Upon stress, Ratan root growth was drastically affected, and the leaves' relative water content was significantly reduced. On the other hand, the increase on ABA and *β*-ODAP levels was significantly higher in Pusa-24, suggesting an upregulation in *β*-ODAP biosynthetic genes. The differential regulation of these genes suggested an altered physiological balance that helped the plant to interact with its environment, and enhanced stress tolerance [113].

Disclosing the influence of the environment on grass pea resilience will provide useful information to breeders focused on improving crop yields and quality, as well as to farmers facing climate change. This information will be useful to understand which are the best breeding approaches for climate change and the best cropping systems for a more resilient production.

## 5. Breeding and Agroecological Transition on Grass Pea Production Systems: Strategies toward Improved Resilience

Breeding for crop productivity and climate resilience is a "big aim" for crop improvement [114]. To achieve this is necessary to understand crops responses under different limiting factors that are becoming more frequent threats with climate change [115]. The

most important limiting factors, hampering crop productivity, are pests and disease outbreaks, high temperatures, drought, flood, soil low fertility, and salinity [116]. Therefore, the need for yield resilience, multi-stress resistance, and hardy crops is paramount. The essential more efficient precision breeding to attain this type of crop relies on a proper understanding of the mechanisms underlying the stress responsiveness of crop species with adaptation traits [115,117].

Simultaneously, the adoption of more resilient agronomic production strategies, such as intercropping, is desirable since these sustainable practices can improve resource use efficiency, thereby facilitating low-input agriculture [118]. Grass pea has tremendous potential as a source of stress-tolerance/resistance and adaptation genes, thus it may be considered a resilient gene donor plant for general crop improvement under climate change conditions, apart from being by itself an interesting protein-rich crop [47,119]. Exploiting this potential through advances in breeding technologies, combined with improved agronomic approaches, is mandatory to enhance response to climate change challenges.

### 5.1. Approaches for Diversity Creation and Increase Genetic Gains

Advanced breeding tools or technologies for driving genetic gains in climate-vulnerable environments are becoming more available to researchers, enabling them to progress faster on the development of climate-resilient crops. The advance in techniques that increase the diversity available for breeders, such as interspecific hybridization and gene editing, or increase genetic gains, through the reduction of generation time by "speed breeding", and the increased efficiency in breeding targets identification via genomics, will bring up new opportunities for breeders in a diversity of crops [3].

When compared to other legume crops, limited research efforts have been devoted to the genetic improvement of grass pea [16], leading to scarcity of genomic resources and precision breeding tools, which has delayed genetic gain increases especially in climate-vulnerable environments.

#### 5.1.1. Diversity Creation

Historically, the main objective of grass pea breeding was focused on yield improvement [16,26]. This has later on evolved to the development of improved varieties with low $\beta$-ODAP content and, in a third phase, several varieties and lines were developed combining low $\beta$-ODAP (<0.1%) content with high yield potential (up to 1.5 tons/ha) and resistance to a variety of biotic and abiotic stresses [26]. Released grass pea varieties with low $\beta$-ODAP content were deeply reviewed elsewhere [18]. More recently, researchers diversified further their breeding objectives, considering not only yield stability [30], but also seed protein quality, and exploiting the non-neurotoxic potentials of $\beta$-ODAP [16,46]. With the new challenges raised by climate change, grass pea breeding needs now to address an increased variability of stresses, ensuring the development of multiple stress-resistant varieties that withstand drought, flood, heat, and a diversity of diseases or pests and, simultaneously, provide yield and quality stability in uncertain environmental growing condition.

Variability is the basis of any breeding program. Although there is considerable variability in grass pea germplasm around the world [44], this crop, due to its underused, shows also a high risk of suffering from genetic erosion [75]. The risk of genetic erosion of crops together with the treats of climate change impose an urgent need to explore wild genetic diversity [120]. Indeed, to overcome the potential narrow genetic base and mainly, for joining adaptive traits that are not found together in nature, some conventional grass pea breeding programs adopted an interspecific hybridization strategy with the introduction of desirable traits from wild related *Lathyrus* species [27,121]. Wild species related to crops (crop wild relatives, or CWR) can increase the adaptive capacity of agricultural systems. They represent a large pool of genetic diversity from which new allelic variation required in breeding programs can be found [120]. Detailed knowledge on the closest relatives and their origin are important information in the breeding process [122]. Besides the high variability within the *L. sativus* primary gene pool, there is potential for exploitation of

related species in grass pea breeding. Heywood et al. [123] extended the *L. sativus* secondary gene pool to include *L. chrysanthus*, *L. gorgoni*, *L. marmoratus*, *L. pseudocicera*, *L. amphicarpos*, *L. blepharicarpus*, *L. chloranthus*, *L. cicera*, *L. hierosolymitanus* and *L. hirsutus*. The remaining species of the genus are considered members of the tertiary gene pool [47]. Significant successes have been reported on the introduction of traits from CWR into crop species, mainly to overcome biotic stresses [124]. This is presently particularly interesting since the range of many plant pathogens is predicted to shift with the changing climate and, therefore, many areas of the world may experience disease outbreaks not previously faced and for which there is no available ready-to-use resistance [15].

In the *Lathyrus* genus, the interest in experimental interspecific hybridization was shown in sweet pea (*L. odoratus*) as early as 1916 [27]. Successful interspecific hybridizations involving *L. sativus* have only been reported with two species, *L. amphicarpos* and *L. cicera*. This represents a limitation that hampers the use of this technique widely [125–127], but endeavours already good perspectives for traits, such as resistance to broomrape infection which appeared as a major limiting factor for grass pea production in Mediterranean and West Asian countries [30,128,129].

To overcome the lack of natural diversity in particular traits that could not be solved through interspecific hybridization limitations, new breeding technologies such as gene editing have emerged to help create new diversity [130–132]. Gene editing is based on the use of engineered nucleases and cellular DNA repair pathways to make precise, targeted changes to the genome of an organism [133]. The development of the CRISPR/Cas9-mediated gene-editing technology has broadened the options to modify genes through the addition or deletion of genetic material in an efficient manner. A successful application of genome editing depends on the possibility to transform and regenerate an entire plant [133]. Legume species are well known to be recalcitrant in terms of regeneration and grass pea, in particular, has quite problematic somatic embryogenesis or organogenesis [16,134]. Thus, the availability of an efficient and reproducible regeneration protocol for grass pea is of paramount importance. A regeneration protocol of fertile plants from meristematic tissues seems to be established for *L. sativus* [135,136]. The development of an efficient CRISPR-cas9/regeneration protocol package for grass pea would be particularly interesting for the reduction/removal of $\beta$-ODAP content through its application on key enzymes in its biosynthetic pathway [137,138].

### 5.1.2. Increase Genetic Gain

Until the moment, the crop plants genetic gain increase rate has been slow, sometimes due to the long generation time of crop plants [139]. In addition, also the reduced breeding efficiency due to the lack of knowledge on the interesting traits genetic basis [140], or the lack of high-throughput phenotyping approaches that could efficiently analyse the high number of samples routinely handled in breeding programs [141,142] have contributed to this slow rate.

"Speed-breeding" techniques are light-based techniques that accelerate photosynthesis and flowering by optimization of light quality, intensity, day length, and temperature control [143]. This allows early seed harvest, with a greatly short generation time, and accelerates breeding and research programs [143]. As a legume example, speed breeding protocols have been used to achieve up to 6 generations per year for pea [141,144]. In grass pea, Barpete et al. [145], developed an accelerated flowering protocol that allowed obtaining 4.33 generations per year, decreasing the time needed from $F_1$ to the $F_7$ generation, to obtain homozygous lines.

There is also potential to accelerate even further the rate of crop improvement by integrating speed breeding with other modern breeding technologies, including high-throughput phenotyping and genotyping, marker-assisted selection, or even genomic prediction [139,143].

Breeding programs need to phenotype a large number of genotypes for complex traits, like quality-related such as phenolic compounds, protein content and essential amino acids

to select the best ones for the next breeding and selection cycles. For some quality traits, high throughput selection tools, such as spectroscopic tools [146], are now becoming available to implement routinely quality objectives in breeding programs [92]. Near-infrared spectroscopy (NIR) and Fourier Transform Infrared spectroscopy (FT-IR) are well-established techniques for determining components of foods. Contrastingly to standard chemical analyses, that are normally laborious and time-consuming, sample preparation for NIR or FT-IR is minimal and with a short measurement time, making them suitable for breeding. NIR and FT-IR spectroscopic methods have been independently developed to determine nutritional components such as protein content and structure/digestibility, moisture, fat, ash, starch, dietary fibre, phytate, essential amino acids or several minerals, sensory traits and cooking time in some legume species other than grass pea [146–154]. FT-IR spectroscopic patterns of a collection of 100 grass pea accessions have already been analysed by multivariate analysis to contribute to the development of an innovative classification approach that differentiated among five grain legume species allowing the identification of outliers in all the species. These accessions might in the future be associated with a specific biochemical composition to develop prediction models to introduce in breeding program [155]. The development of a reliable spectroscopic prediction model for $\beta$-ODAP in grass pea would represent an interesting advance for quality breeding. Presently it is already possible by HPLC-MS/MS to independently quantify both ODAP isomers (the nontoxic $\alpha$ and the toxic $\beta$) instead of the total amount of ODAP [156], but if a more cost and time effective, non-destructive (whole seed) spectroscopic approach would be available, it would represent a breakthrough in precision quality breeding.

Quicker progress on crop improvement rate can also be attained by the efficiency increase in breeding targets identification through genomics. Once a trait is associated with one or several molecular markers (or in the best of the options with its functional genes) at DNA level, plants can be selected early on their growth stage, allowing a faster and more efficient breeding process [82].

Although there has been an encouraging recent growth of available genomic information in the *Lathyrus* genus, these resources are still modest when compared with other legume crops [47]. Advances have been achieved with molecular markers developed for *L. sativus* allowing the detection of diversity and variation among and within species [16]. PCR_based molecular markers, such as Simple Sequence Repeat (SSR) implementation in grass pea have remarkably improved the efficiency in distinguishing between different *L. sativus* accessions and in assessing the within-species genetic variability [10,17,47,76].

Additionally, molecular markers can be used to determine the number, position, and individual effects of genes/quantitative trait loci (QTLs) that control interesting traits, such as disease or pest resistance, $\beta$-ODAP and protein concentrations, and other characteristics of agronomic importance, via, for instance, genetic linkage mapping and association mapping [12,83].

The application of next-generation sequencing (NGS) to grass pea has provided a means to develop further the repertoire of genomic resources for this species. Different NGS platforms were used to generate and develop not only SSR markers but also single nucleotide polymorphisms (SNPs) that facilitated the construction of high-resolution maps for comparative and QTL mapping in grass pea [10,16,76,157]. In addition, genotyping-by-sequencing (GBS), a targeted marker technique that has been developed using NGS, was frequently used in genome-wide association studies (GWAS) in grass pea [77,83] (details on associated genomic regions in the previous section "Grass pea biotic stresses resistance"). In grass pea, a reference draft genome is now available which reveals an approximate size of 6.3 Gb [158]. Nevertheless, this draft genome is still not fully assembled making the interpretation of GWAS results a considerable challenge [83]. This may however be overcome through comparative mapping by resorting to the SNP markers' genomic positions retrieved from the phylogenetically related pea's reference genome v1a [159]. In order to increase grass pea breeding efficiency also at agronomic/adaptation or quality related traits level, all complex in their nature, more genetic studies of this type are still needed.

*5.2. Transition to More Sustainable and Resilient Production Systems*

The growth and yield of a crop genotype will be affected by all the components of an agronomic package: sowing, spacing, weeding, soil fertility enhancement, and soil moisture control. Changes in any of these components will have major consequences on the levels of stress, and their effects on the crop [160]. Climate change can be considered a serious risk factor due to the pressure that could create in the agronomic system. As response to this pressure, some agricultural practices, such as delaying sowing dates, crop rotation, and intercropping may be implemented [160]. As an example, one of such practices, the association of annual legumes with cereals, can exploit plant functional diversity to raise crop yields, yield stability, and/or crop quality, while simultaneously enhancing ecosystem services and reducing adverse environmental impacts [6].

Intercropping is one of the agronomic techniques that serve as a base pillar to sustainable agriculture. It consists of cultivating two or more crops in the same space allowing to increase productivity per unit area of land, a better utilization of resources, minimizing the risks, reducing weed competition, and stabilizing yield [161]. Intercrop combining different species such as legumes with cereals, enhances the resistance against different pathogens and weeds with a positive effect on crop productivity [4]. Additionally, the use of legumes, improves soil fertility, due to their capability to fix biological nitrogen, and conservation against erosion through greater ground cover when compared with monoculture [67,161].

Grass pea has been cultivated in different regions as part of a diversity of non-intensive agricultural systems, as a resilient and low-input crop. In Ethiopia, grass pea is commonly cultivated on heavy clay, and iron-rich soils (mostly vertisols). In here, planting as a sole crop is done in late August to early September, and as such, the crop is supposed to take advantage of the residual soil moisture [46]. In Bangladesh and West Bengal, India, grass pea is mostly grown as a relay crop in low-lying areas in *Aman* rice fields. Its broadcast sowing with high moisture in standing rice field will take place 3–4 weeks before harvest [32]. In Nepal, its cultivation is mostly restricted to marginal areas like waterlogged, lowland rice areas where farmers usually cannot take other winter crops like wheat, oilseeds, or other legumes. Here no additional chemical fertilizer or insecticide is used in its cultivation [32]. Indeed, a low input system is the most common wherever you grow grass pea [162,163]. That is also the case, for instance of Portugal, in southern Europe where grass pea is cultivated as a sole Autumn-Winter crop for grain production, integrated into crop rotation, mainly alternating with cereals or in intercropping systems with olive (*Olea europea*), chickpea (*Cicer arietium*) or faba bean (*Vicia faba*) [38]. As a dual-purpose crop, with great agronomic potential as a grain and forage legume, grass pea provides good opportunities to diversify existing cereal-based cropping systems [5]. However, not many reports to the use of grass pea in intercropping systems are available. Atis and Acikalin [164] compared grass pea (GP) and wheat (W) pure stands as well as their mixtures for forage yield and quality, and to estimate the effect of the species competition in the intercropping systems. They concluded that 60% GP + 40% W mixture gave the best results in terms of yield and quality. Also, Rhaman et al. [165] evaluated a mixture of grass pea and mustard to evaluate the solar radiation as radiation use-efficiency and concluded that both crops are well compatible in intercrop association. Grass pea has been identified also as a good alternative to summer fallow if used as a ground cover, green manure or forage crop in several regions, as in the American Great Plains [166,167].

## 6. Conclusions

Climate change will affect agriculture differently, depending on both, the region and the crop. The most significant stresses will be due to variable rainfall, reduced water availability, temperature raising, frequent occurrence of extreme events, and the spread of pest and disease outbreaks. A crop yield decline is expected, thus alternative varieties or new crops will be required to ensure a stable food supply. Challenges will be set mainly in crop traditional growing regions and to overcome them the development of varieties displaying resilience to prevalent biotic and abiotic stresses has gained new strength.

Grass pea is a hardish crop and a strong alternative candidate under these circumstances since it presents a set of important stress resilient/adaptive traits. Grass pea is simultaneously a highly nutritive and healthy food crop, with one only compositional drawback, the presence of an antinutrient, *β*-ODAP, considered the cause of lathyrism. Nevertheless, the hardy nature of this crop is also suggested to be related to the presence of *β*-ODAP within the plant.

The identification of grass pea sources of the resilient/adaptive traits is essential for breeding, as well as the understanding of their genetic control that will allow pinpointing promising genomic targets for the development of molecular tools to assist a multi-stress resistance and quality precision breeding in this species.

For the needed simultaneous targeting of multiple complex traits, we propose a multidisciplinary approach starting by an high throughput phenotyping in different stages of a growing cycle, considering yield parameters but also quality traits that may rely on cost-efficient phenotyping tools, such as FT-IR, that at the end will contribute to accelerating genetic gain under climate changes. Moreover, quicker progress on crop improvement rate can also be attained by the efficiency increase in breeding targets identification through genomics. Increasing genomic knowledge and tools are becoming available in grass pea, such as PCR based molecular markers, QTLs, linkage and association mapping studies allowing clarifying the putative candidate genes and mechanisms underlying interesting traits.

On the other hand, agricultural practices such as intercropping are starting to be considered pillars of sustainable agriculture. Grass pea is a model crop for sustainable agricultural systems due to their resilience to several abiotic and biotic stresses and by being a legume species with low required inputs (such as water or N fertilization) and great versatility to be included in crop rotations or intercropping, may be an important answer to climate challenges.

Grass pea is a suitable crop to address an integrative strategy to tackle climate change impacts coupling agronomic-quality-based improvements with the implementation of more sustainable agricultural techniques and variety-breeding-based strategies based on the set of resilient grass pea traits.

**Author Contributions:** Writing—original draft, L.G.; writing—review and editing, M.R.B., D.R. and M.C.V.P.; funding acquisition, M.C.V.P. All authors have read and agreed to the published version of the manuscript.

**Funding:** This work was supported by the Fundação Para a Ciência e Tecnologia through the grant SFRH/BD/124094/2016 (LG), the R&D Research Unit GREEN-IT—Bioresources for Sustainability (UIDB/04551/2020, UIDP/04551/2020) and the LS4FUTURE Associated Laboratory (LA/P/0087/2020), by Spanish Research Agency (AEI) project PID2020-11468RB-100 and by the European Union Horizon 2020 Research and Innovation Programme under Grant no 101000383 (DIVINFOOD).

**Conflicts of Interest:** The authors declare no conflict of interest.

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
