# Peer review of "Grass Pea (Lathyrus sativus L.)—A Sustainable and Resilient Answer to Climate Challenges"

_agronomy, doi:10.3390/agronomy12061324_

Round 1

Reviewer 1 Report

As promising source of traits for adaptation to climate change, grass pea is indeed a hardish crop with reported tolerance to extreme temperatures, drought, flooding, and salinity being able to grow successfully in warm climates, and marginal and nutrient-deficient soils, delivering reasonably good yield despite unfavourable growing conditions,  with resistant to many diseases and pests compared to other legume crops.

The authors thoroughly summarized the important publications up-to-date, and made this wonderful review available to readers who will find a very promising new legume crop for dealing with climate challenges in cropping systems globally.

One of the major difference between grass pea and other legume crops is its contents of ODAP, including β-ODAP and α-ODAP. If possible, α-ODAP should be somewhat described when β-ODAP is mentioned.

Reviewer 2 Report

Dear authors, the manuscript is very well written. The individual scientific questions and solutions are presented in depth and with routine. Literary sources completely corresponding to the considered topic were used. I have only one remark. The conclusion presented at the end of the article is very long. In my opinion, it should be presented in a tighter form. Authors should not be cited in this section. It would be better to remove the citations.

Reviewer 3 Report

The MS “Grass Pea (Lathyrus sativus)- a sustainable and resilient answer to climate challenges” is interesting. The information compiled in the MS will be helpful for the scientific community to explore the L. sativus as an alternate pulse crop for challenging environments. Manuscript is well written and presented in good manner; the contents of manuscript are presented appropriately. Below I have pointed some issues which authors can consider while revising the MS.

Line#35-38. Check the sentence.

Line#52 Write human and animal feed.

Line#104-106. Please double-check the numerical order of citations and make necessary changes throughout the manuscript.

Line#229-230. The citation Vaz Patto and collaborators should be changed to Vaz Patto et al. [71]. The same can be corrected throughout the manuscript. FYI ‘et al.’ means “and others”, so there is no need to write and collaborators after the name of first author.

Line#456. Bromrape should be changed to broomrape.

Line#633. PCR based.

#There is room for improvement in the MS by incorporating information from the following recent publications.

Divya Rathi, Jitendra Kumar Verma, Akanksha Pareek, Subhra Chakraborty, Niranjan Chakraborty. Dissection of grasspea (Lathyrus sativus L.) root exoproteome reveals critical insights and novel proteins. Plant Science: An International Journal of Experimental Plant Biology 2022, 316: 111161

Moshe Goldsmith, Shiri Barad, Yoav Peleg, Shira Albeck, Orly Dym, Alexander Brandis, Tevie Mehlman, Ziv Reich. The identification and characterization of an oxalyl-CoA synthetase from grass pea (Lathyrus sativus L.). RSC chemical biology 2022 March 9, 3 (3): 320-333

Moshe Goldsmith, Shiri Barad, Maor Knafo, Alon Savidor, Shifra Ben-Dor, Alexander Brandis, Tevie Mehlman, Yoav Peleg, Shira Albeck, Orly Dym, Efrat Ben-Zeev, Ranjit S Barbole, Asaph Aharoni, Ziv Reich. Identification and characterization of the key enzyme in the biosynthesis of the neurotoxin β-ODAP in grass pea. Journal of Biological Chemistry 2022 March 7,: 101806

Aveek Samanta, Saptadipa Banerjee, Tilak Raj Maity, Jangala Jahnavi, Siraj Datta. Towards establishment of a plant-based model to assess the novel anti-cancerous lead molecule(s): An in silico, in vivo and in vitro assessment of some potential anti-cancerous drugs on Lathyrus sativus L Protoplasma 2022 February 23

Anjali Verma, Nidhi Nidhi, Gazaldeep Kaur, Shrikant Mantri, Tilak Raj Sharma, Ajay Kumar Pandey, Pramod Kaitheri Kandoth. Contrasting β-ODAP content correlates with stress gene expression in Lathyrus cultivars Physiologia Plantarum 2022, 174 (1): e13616

Hai-Yan Kong, Hao Zhu, Rui Zhou, Nudrat Aisha Akram, Yi-Bo Wang, Cheng-Jing Jiao, You-Cai Xiong. Role of abscisic acid in modulating drought acclimation, agronomic characteristics, and β-N-Oxalyl-L-α, β-diaminopropionic acid (β-ODAP) accumulation in grass pea (Lathyrus sativus L.). October 23, 2021: Journal of the Science of Food and Agriculture

# If possible, authors can add one figure and one illustration to make the MS more comprehensive.

#In section 5.1.1 the authors can add a paragraph about utilization of related species from the secondary and tertiary gene pools.

# At the same time authors are expected to improve the grammatical errors, and spelling mistakes.

With best wishes!
